# Distinct spin and orbital dynamics in Sr$_2$RuO$_4$

H. Suzuki [1,2,3,15] ✉, L. Wang [1,15], J. Bertinshaw[1], H. U. R. Strand [4,5], S. Käser[1,6], M. Krautloher [1], Z. Yang[1], N. Wentzell [7], O. Parcollet[7,8], F. Jerzembeck [9], N. Kikugawa [10], A. P. Mackenzie [9], A. Georges[7,11,12,13], P. Hansmann [1,6,9], H. Gretarsson[1,14] ✉ & B. Keimer [1] ✉

The unconventional superconductor Sr$_2$RuO$_4$ has long served as a benchmark for theories of correlated-electron materials. The determination of the superconducting pairing mechanism requires detailed experimental information on collective bosonic excitations as potential mediators of Cooper pairing. We have used Ru $L_3$-edge resonant inelastic x-ray scattering to obtain comprehensive maps of the electronic excitations of Sr$_2$RuO$_4$ over the entire Brillouin zone. We observe multiple branches of dispersive spin and orbital excitations associated with distinctly different energy scales. The spin and orbital dynamical response functions calculated within the dynamical mean-field theory are in excellent agreement with the experimental data. Our results highlight the Hund metal nature of Sr$_2$RuO$_4$ and provide key information for the understanding of its unconventional superconductivity.

Conduction electrons in quantum materials form itinerant quasiparticles that propagate coherently over mesoscopic length scales, while being renormalized by local interactions akin to those in atomic physics. This dichotomy spawns a large variety of collective quantum phenomena and remains one of the major challenges of modern condensed matter physics, as epitomized by the Hubbard model describing electrons on a lattice with a single orbital per site, which has defied a complete solution until today—60 years after it was first introduced. Coulomb repulsion of opposite-spin electrons residing on the same site drives the electron system toward a Mott insulating state and induces antiferromagnetic spin correlations, which have been invoked as a mediator of Cooper pairing in unconventional

superconductors such as cuprates[1] and nickelates[2]. Whereas bona-fide realizations of the Hubbard model are rare, studies on transition metal oxides[3] and iron-based superconductors[4] have led to the realization that atomic Hund's rule interactions among conduction electrons with multiple active $d$-orbitals are a source of strong electron correlations. In principle, treating spin and orbital correlations on an equal footing further increases the challenge in the theoretical description of the interacting electron system. However, recent dynamical mean-field theory (DMFT) studies of the broad family of "Hund metals"[5] have suggested that the Hund's rule interactions drive a large-scale differentiation of spin and orbital screening energies[6,7]. Indirect manifestations of this "spin-orbital separation" include the formation of local

[1]Max-Planck-Institut für Festkörperforschung, Heisenbergstraße 1, D-70569 Stuttgart, Germany. [2]Frontier Research Institute for Interdisciplinary Sciences, Tohoku University, Sendai 980-8578, Japan. [3]Institute of Multidisciplinary Research for Advanced Materials (IMRAM), Tohoku University, Sendai 980-8578, Japan. [4]School of Science and Technology, Örebro University, Fakultetsgatan 1, SE-701 82 Örebro, Sweden. [5]Institute for Molecules and Materials, Radboud University, 6525 AJ Nijmegen, the Netherlands. [6]Department of Physics, Friedrich-Alexander-University (FAU) of Erlangen-Nürnberg, 91058 Erlangen, Germany. [7]Center for Computational Quantum Physics, Flatiron Institute, Simons Foundation, 162 5th Avenue, New York 10010, USA. [8]Université Paris-Saclay, CNRS, CEA, Institut de physique théorique, 91191 Gif-sur-Yvette, France. [9]Max Planck Institute for Chemical Physics of Solids, Nöthnitzer Straße 40, 01187 Dresden, Germany. [10]National Institute for Materials Science, Tsukuba, Ibaraki 305-0003, Japan. [11]Collège de France, 11 place Marcelin Berthelot, 75005 Paris, France. [12]Centre de Physique Théorique (CPHT), CNRS, Ecole Polytechnique, IP Paris, 91128 Palaiseau, France. [13]Department of Quantum Matter Physics, University of Geneva, 24 Quai Ernest-Ansermet, 1211 Geneva 4, Switzerland. [14]Deutsches Elektronen-Synchrotron DESY, Notkestraße 85, D-22607 Hamburg, Germany. [15]These authors contributed equally: H. Suzuki, L. Wang. ✉e-mail: hakuto.suzuki@tohoku.ac.jp; hlynur.gretarsson@desy.de; b.keimer@fkf.mpg.de

magnetic moments and the anomalously low onset temperature of the coherent Fermi-liquid state in iron-based superconductors.

Here we report a direct spectroscopic fingerprint of spin-orbital separation in the archetypical Hund metal $Sr_2RuO_4$[8], which has been the subject of many years of study in view of the textbook Fermi-liquid transport properties[9] and unconventional superconducting state[10] that develop upon cooling below the coherence temperature $T_{coh} \sim 25$ K and critical temperature $T_c = 1.5$ K, respectively. Recent precision experiments in the superconducting state[11–13] have cast doubt on the previously advocated spin-triplet pairing scenario[14,15], thus revitalizing the order parameter debate[16] and the search for an in-depth understanding of the Fermi-liquid normal state and collective bosonic fluctuations relevant to superconductivity. The lattice structure of $Sr_2RuO_4$ is built up of $RuO_6$ octahedra in a square-planar arrangement, and its Fermi surface comprises three bands originating from the $d$-orbital manifold of $Ru^{4+}$ ions in the octahedral crystal field. Owing to the availability of exceptionally clean single crystals, the electronic quasiparticle properties in these bands are very well known and clearly indicate strong electronic correlations. The temperature dependence of the Seebeck coefficient[17,18] suggests a separation between energy scales associated with the onset of coherence of spin and orbital degrees of freedom, consistent with the notion that these correlations are governed by Hund's rules[19,20]. Inelastic neutron scattering (INS) studies of $Sr_2RuO_4$[21–24] have revealed low-energy incommensurate spin fluctuations (ISFs) at the in-plane wavevectors $\mathbf{q}_{ISF} = (\pm 0.3, \pm 0.3)$ and along a square-shaped ridge connecting them (Fig. 1a, inset). However, the INS spectra are limited to energies below ~50 meV and do not yield separate information on orbital excitations.

We have used Ru $L_3$-edge (2838 eV) resonant inelastic x-ray scattering (RIXS) to obtain spectroscopic maps of spin and orbital fluctuations over a wide range of energy and momenta. Whereas the spin excitations are almost completely confined to energies below ~200 meV, significant orbital fluctuations only appear at higher energies and extend up to ~1 eV, thus directly confirming the theoretically predicted spin-orbital separation. The RIXS spectra disagree starkly with predictions based on the standard random phase approximation (RPA), which do not capture the distinct energy scales of spin and orbital correlations, but are in excellent agreement with calculations in the framework of DMFT, which takes into account vertex corrections. Our results thus demonstrate the key role of Hund's-rule interactions in inducing electron correlations in multiband metals, and highlight the capability of current many-body theory to accurately compute two-particle correlation functions. They also shed light on the nature of potential pairing bosons for unconventional superconductivity in $Sr_2RuO_4$.

## Results and discussion

Figure 1a shows the crystal structure of $Sr_2RuO_4$ and the scattering geometry for the RIXS experiment. The incident x-ray photons were $\pi$-polarized, and the scattered photons with both $\sigma$ and $\pi$ polarizations were collected at the scattering angle of 90 degrees. In this geometry, the polarizations of the incident and outgoing photons are always perpendicular, selectively enhancing magnetic responses from the spin and orbital excitations while suppressing the charge response. Given the layered crystal structure of $Sr_2RuO_4$, we express the momentum transfer using the in-plane component $\mathbf{q}$, which is scanned by changing the sample angle $\theta$. We studied two paths in the reciprocal space, $\mathbf{q} = (H, 0)$ and $(H, H)$, by fixing the azimuthal angle $\phi$ at $0°$ and $-45°$, respectively. These paths cross the ridge and the peak of the low-energy ISFs (inset). The measurements were performed at $T = 25$ K, in the FL regime of the normal state.

In Fig. 1b, we show the Ru $L_3$ RIXS spectra along the two directions. Multiple peak structures are readily identified. The main feature A is composed of multiple peaks which extend up to ~1 eV. These peaks are assigned to spin and orbital excitations within the $t_{2g}$ orbitals. In addition, a weakly-dispersive feature B is identified at ~3 eV (blue circles). As this energy corresponds to the splitting of the transitions to the unoccupied $4d$ $t_{2g}$ and $e_g$ orbitals in the Ru $L_3$ x-ray absorption spectrum (Supplementary Fig. 1a), the feature B is readily assigned to the crystal field transitions to the $t_{2g}^3 e_g^1$ electron configurations. We note here that its intensity is maximal close to the $\mathbf{q} = (0, 0)$ point along the two directions.

To visualize the characteristics of the RIXS spectra, we show in Fig. 1c a colormap of the RIXS intensity. The main feature A is composed of multiple dispersions. Its low-energy tail exhibits downward dispersion toward its local minima at $\mathbf{q} = (-0.3, 0)$ and $(-0.7, 0)$ along the $(H, 0)$ direction and at $\mathbf{q}_{ISF} = (-0.3, -0.3)$ along the $(H, H)$ direction (white triangles). These $\mathbf{q}$ vectors are in excellent agreement with those of the ridges and ISFs identified in the previous INS studies[23,24]. However, the information from the INS data is limited to the low-energy region below ~0.1 eV, whereas the full access to a large energy window in the present RIXS experiment provides comprehensive information on the ISFs. The relative intensity of spin excitations and the location of the ISF are determined by the nesting conditions between the multiple Fermi surface sheets of $Sr_2RuO_4$. It is well established that the nesting between the $\alpha$ and $\beta$ sheets with $\mathbf{q}_{ISF}$, as

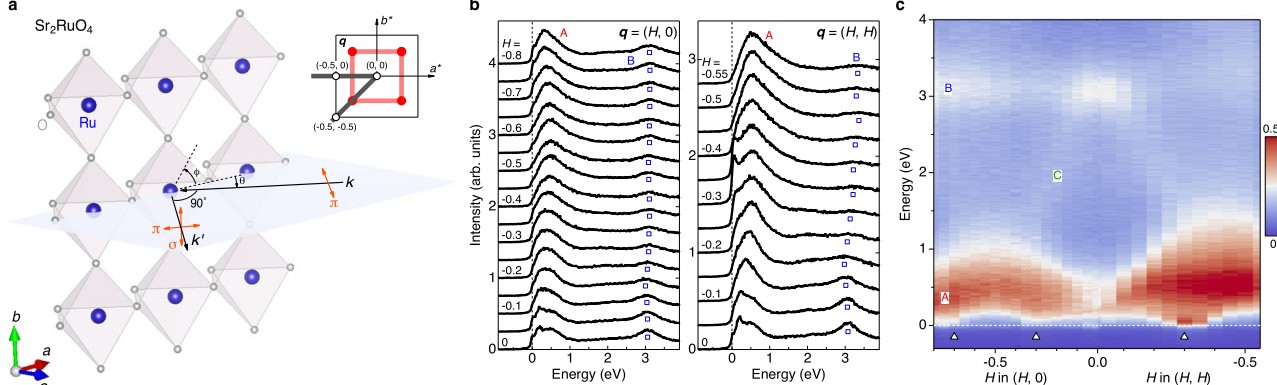

**Fig. 1 | Ru $L_3$ RIXS of $Sr_2RuO_4$. a** Crystal structure of $Sr_2RuO_4$ and scattering geometry for the resonant inelastic x-ray scattering (RIXS) experiment. Incident x-ray photons with momentum $\mathbf{k}$ are linearly $\pi$-polarized and the polarization of the scattered photons with momentum $\mathbf{k}'$ is not analyzed. The scattering angle is fixed at $90°$ and the in-plane momentum transfer $\mathbf{q}$ is scanned by rotating the sample angle $\theta$. The azimuthal angle $\phi$ is used to change the measurement paths in the reciprocal space (gray lines, inset). Red circles and lines in the inset are the schematics of the low-energy incommensurate spin fluctuations (ISFs). **b** Ru $L_3$ RIXS spectra along the $\mathbf{q} = (H, 0)$ and $(H, H)$ directions. Blue squares indicate the peak positions of crystal field transitions. **c** Colormap of RIXS intensity. The positions of ISFs are indicated by open triangles.

demonstrated by angle-resolved photoemission measurements[25], drives the incommensurate spin fluctuations with $q_{ISF}$[26]. In contrast, the nesting is only partial along the $(H, 0)$ direction, resulting in the reduced intensity of the spin fluctuations along this direction.

The colormap also reveals an additional broad dispersive feature C at high energy ($\geq$1 eV). It emanates from the top of the feature A at $\mathbf{q} = (-0.5, 0)$ and $(-0.5, -0.5)$ and merges with the feature B at the $(0, 0)$ point, generating the intensity maximum. At the high-symmetry $(0, 0)$ point, the $dd$ excitations to the $d_{x^2-y^2}$ and $d_{3z^2-r^2}$ orbitals remain localized and are almost degenerate in energy under the small tetragonal distortion of the $RuO_6$ octahedra. At finite in-plane $\mathbf{q}$'s, the excitations to the planar $d_{x^2-y^2}$ orbitals show energy dispersion due to large overlap integrals with the planar O $2p$ orbitals, while those to the out-of-plane $d_{3z^2-r^2}$ orbitals have little in-plane dispersion due to the small overlap integrals. The nondispersive feature B and dispersive feature C are thus primarily ascribed to the transitions to the $d_{3z^2-r^2}$ and $d_{x^2-y^2}$ orbitals, respectively. The dispersion of the orbital fluctuations originates from the large bandwidth of $Sr_2RuO_4$ with a tetragonal crystal structure. In contrast, the nondispersive orbital excitations observed in orthorhombic $Ca_2RuO_4$[27] and $Ca_3Ru_2O_7$[28] indicate that the local $dd$ excitations cannot freely move to the neighboring sites, as the rotation of the $RuO_6$ octahedra significantly reduces the hopping integrals.

Having identified multiple branches of spin-orbital excitations in $Sr_2RuO_4$, we now scrutinize the low-energy excitations within the $t_{2g}$ orbitals (feature A). Figure 2a shows an expanded plot of the RIXS spectra below 0.8 eV. The broad global peak maxima (red circles) disperse from 0.2 eV at the zone center $\mathbf{q} = (0, 0)$, where they are most sharply peaked, to the maximal energy at the zone boundary, $\mathbf{q} = (0, -0.5)$ and $(-0.5, -0.5)$. We ascribe this dispersion to orbital

fluctuations as we will see below. The energy scale of orbital fluctuations near the zone center agrees with that of the O $K$-edge RIXS data[29]. Along the $(H, H)$ direction, the low-energy region contains prominent peaks due to the ISFs around $q_{ISF} = (-0.3, -0.3)$ and subsequent shoulder structures connected to the $(0, 0)$ point, as indicated with black circles (Supplementary Note 2). The quasielastic intensity at the $(0, 0)$ point is significantly weaker than at $q_{ISF}$, consistent with polarized INS data[22]. On the other hand, the spin fluctuation intensity is weaker along the $(H, 0)$ direction, except for the small increase of the quasi-elastic intensity of the ridge scattering around $(-0.3, 0)$ and $(-0.7, 0)$[23,24]. In addition, the spectra close to the $(0, 0)$ point contain a broad high-energy tail peaked around -0.5 eV (Supplementary Note 5).

Figure 2b summarizes the $\mathbf{q}$ dispersions of the observed RIXS features. Here, the dispersion of the orbital fluctuations is defined as the global peak maxima of the RIXS spectra, and that of spin fluctuations as the local maxima of spectral curvature deduced from the second derivative analysis (Supplementary Fig. 2). Along the $(H, 0)$ direction, the orbital fluctuations disperse from -0.2 eV at the $(0, 0)$ point and reach the maximum of -0.5 eV at the $(-0.5, 0)$ point. Along the $(H, H)$ direction, the dispersion is initially steeper and becomes almost flat in the region $H \leq -0.25$ at a higher energy -0.55 eV. The spin excitations have a local minimum of 0.06 eV at $q_{ISF} = (-0.3, -0.3)$ and approach zero energy close to the $(0, 0)$ point. Note here that the spin and orbital fluctuations have distinct energy scales in the entire $\mathbf{q}$ space without a clear signature of mutual crossing. This observation is of crucial importance in testing the validity of different theoretical approaches.

To facilitate a direct connection to the INS results, we show in Fig. 2c an expanded colormap of the RIXS intensity around $q_{ISF}$ and corresponding energy distribution curves with a step size of 0.02 eV.

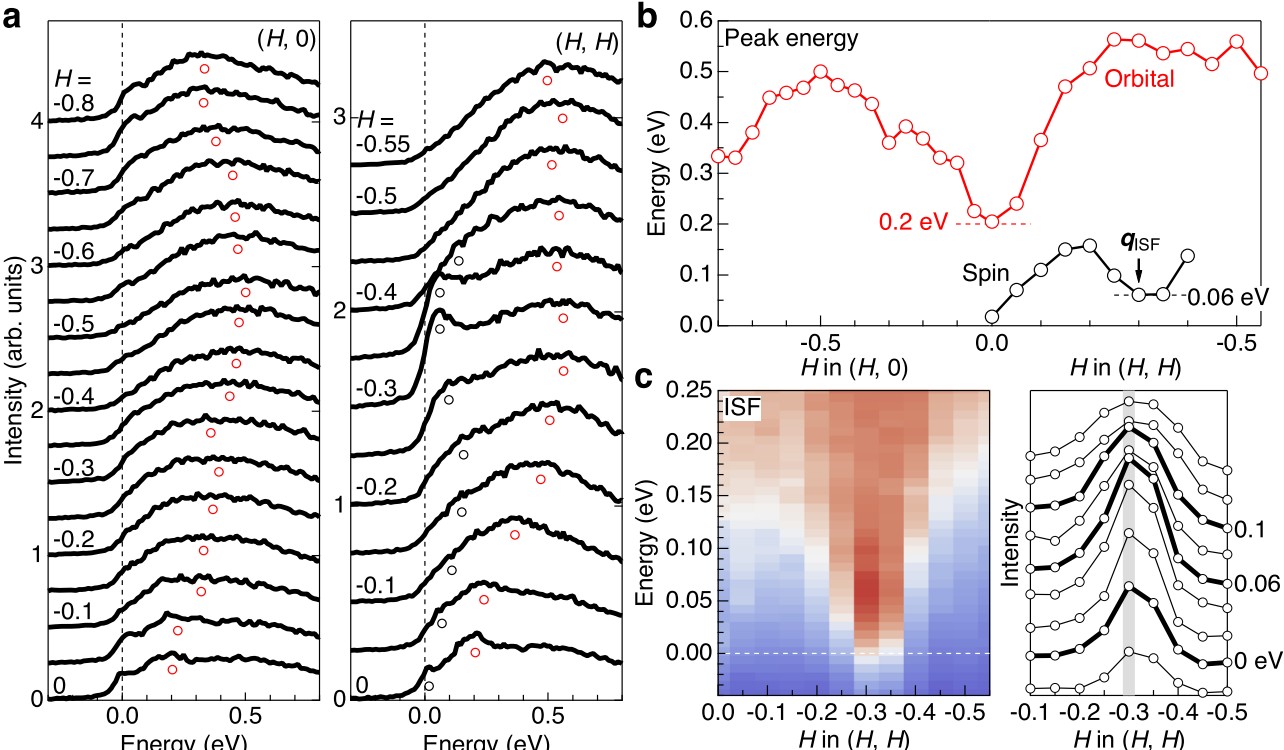

**Fig. 2 | Spin and orbital fluctuations within the $t_{2g}$ orbitals. a** Expanded plots of low-energy RIXS spectra. The global peak maxima corresponding to the orbital fluctuations are indicated by red circles. Along the $(H, H)$ direction, the local maxima and shoulder structures from spin fluctuations are indicated with black circles. **b** Dispersion relations of the spin and orbital fluctuations as a function of the in-plane momentum transfer. The dispersion of the spin fluctuations is defined

as the maxima of local spectral curvature deduced from the second derivative analysis, and that of orbital fluctuations as the global maxima of the original RIXS spectra. The typical error bars associated with the numerical maximum search are smaller than the marker size. **c** Expanded colormap of RIXS intensity around the ISF. The right panel shows momentum distribution curves with a step of 0.02 eV.

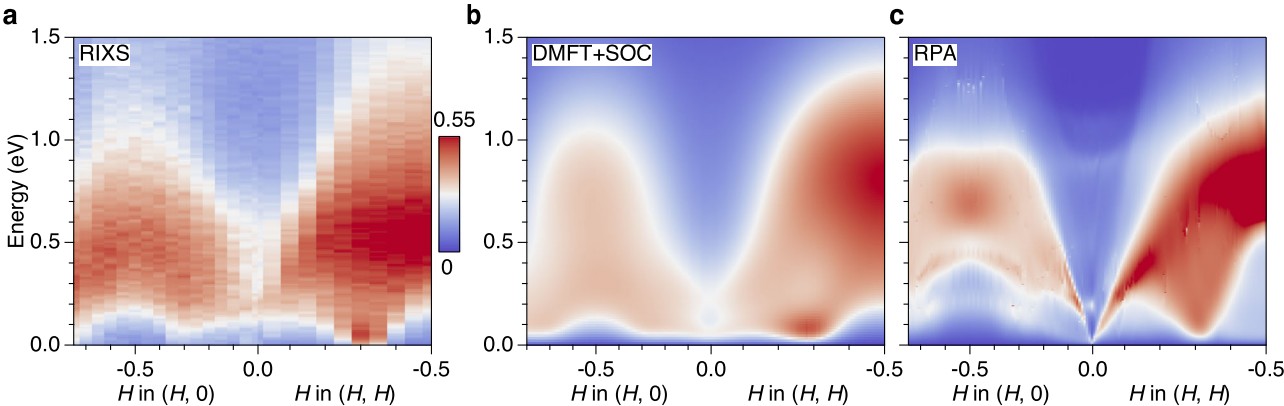

**Fig. 3 | Modelling of RIXS spectra by DMFT+SOC calculations. a** Expanded colormap of the RIXS intensity within the $t_{2g}^4$ electron configurations. **b** Simulation of RIXS intensity based on the spin and orbital susceptibilities calculated by the

dynamical mean-field theory with spin-orbit coupling (DMFT+SOC). **c** Simulation from the susceptibilities calculated with the random phase approximation (RPA).

The RIXS intensity around $\mathbf{q}_{ISF}$ shows a conical shape with an isolated intensity maximum at 0.06 eV. The intensity remains centered at $\mathbf{q}_{ISF}$ up to 0.25 eV, consistent with the vertical INS intensity profile at $\mathbf{q}_{ISF}$ observed below ~0.06 eV[30].

We now interpret the low-energy RIXS data in terms of theoretical spin and orbital susceptibilities $\chi_{S_\mu S_\mu}$ and $\chi_{L_\mu L_\mu}$ ($\mu = x, y, z$), which we computed in DMFT by solving the Bethe-Salpeter equation using the local DMFT particle-hole irreducible vertex within the Ru $4d$-$t_{2g}$ subspace (Supplementary Note 3). We employed the same effective model and interaction parameters that have been established in several previous studies[31–33]. Theoretical RIXS spectra are constructed by combining different components of the spin and orbital susceptibilities with matrix elements for the RIXS cross section (Supplementary Note 4). Figure 3 shows the comparison of the experimental (Fig. 3a) and theoretical DMFT (Fig. 3b) RIXS spectra along the previously defined high-symmetry momentum paths in the Brillouin zone. To highlight the importance of the DMFT dynamical vertex corrections, we also show the perturbative RPA spectra (without dynamical vertex) in Fig. 3c. It is evident that the DMFT spectra excellently capture the overall dispersion and the distribution of momentum and energy dependent maxima of the RIXS data. Specifically, the low-energy intensity is peaked at $\mathbf{q}_{ISF}$ and also extrapolates continuously to the corresponding quasistatic intensity close to zero energy[32]. Moreover, the broader maximum emanates from 0.2 eV at $\mathbf{q} = (0, 0)$ and disperses more steeply along the $(H, H)$ direction. In contrast, the spectral weight distribution in RPA fails to capture the low-energy intensity maximum at $\mathbf{q}_{ISF}$ and yields a spuriously sharp feature that extends from high energy to zero energy around the Γ point.

This difference between DMFT and RPA originates from the distinct behavior of the spin and orbital dynamical responses. Figure 4 shows the intensity plots of theoretical orbital ($LL$) and spin ($SS$) susceptibilities along the $\mathbf{q} = (H, 0)$ and $(H, H)$ directions, obtained by averaging all components $\frac{1}{3}\sum_\mu \chi_{L_\mu L_\mu}$ and $\frac{1}{3}\sum_\mu \chi_{S_\mu S_\mu}$. The vertex corrections in DMFT lead to the clear energy separation of spin and orbital contributions predicted for the Hund metals (left panels). The spin response accounts for almost all the spectral weight at low energies up to ~0.2 eV, and becomes negligible above this scale. The concentrated spectral weight around $\mathbf{q}_{ISF}$ and weak ridge scattering around $\mathbf{q} = (-0.3, 0)$ and $(-0.7, 0)$ excellently reproduce the experimental observations. The orbital response sets in at higher energies > 0.2 eV and shows broad maxima centered around commensurate momenta $\mathbf{q} = (-0.5, 0)$ and $(-0.5, -0.5)$. In RPA (right panels), on the other hand, both the spin- and orbital responses disperse and have spectral weight over the entire energy range. The RIXS data thus provide direct and quantitative evidence for the spin-orbital separation in correlated Hund metals as captured by DMFT. Further improvement between the experiment and theory could

be obtained by a more rigorous treatment of the resonance effect in the RIXS cross section, in particular on the spectral weight maximum of the orbital excitations around $\mathbf{q} = (-0.5, -0.5)$.

It should also be noted that the spin-orbit coupling (SOC) effects have been considered only outside the vertex in our DMFT calculations. Comparison to angle-resolved photoemission experiments[31] and inelastic neutron-scattering[32] have justified this procedure on the level of the single-particle spectra and static magnetic susceptibility. The excellent agreement we find in the present work supports the strategy also for the dynamic susceptibility. This situation is contrasted to the spin-orbital $J$ physics in the Mott insulating counterpart $Ca_2RuO_4$[27] and the cubic $K_2RuCl_6$[34], whose magnetic ground states are determined by the interplay between the ionic $J$ multiplets and the strength of intersite exchange interactions. While the $t_{2g}$ electrons of $Sr_2RuO_4$ carry orbital angular momentum, the finite bandwidth of the itinerant electrons partially quenches the orbital momentum. Nonetheless, the SOC brings about significant modification of the single-particle band structure at certain high-symmetry momenta in the Brillouin zone, when multiple bands are degenerate in energy. It is well known that in $Sr_2RuO_4$ this degeneracy occurs in the diagonal direction in the reciprocal space, which leads to the separation of the Fermi-surface sheets[31,35]. Correspondingly, the effect of SOC on the dynamical susceptibilities is most pronounced in the low-energy spin fluctuations at $\mathbf{q} = (0, 0)$ and at $\mathbf{q}_{ISF}$, while the effect on the orbital fluctuations remains minor (Supplementary Fig. 8).

The current findings also have implications for the microscopic mechanisms of the superconductivity in $Sr_2RuO_4$. As primary candidates of bosonic fluctuations mediating the Cooper pairing, the spin and orbital dynamical susceptibilities enter the Eliashberg equations, which in turn determine the SC order parameter. Our combined RIXS and DMFT+SOC results provide a comprehensive description of the momentum distribution, dispersion relation, and spin-orbit composition of low-energy magnetic excitations, which can serve as crucial input for approximate solutions of the Eliashberg equations. Recent theoretical studies suggest that static (RPA) and dynamic (DMFT) vertex approximations lead to qualitatively different SC ground states[33,36–38]. Although computational challenges prohibit rigorous extrapolation of our theoretical results to low temperatures near $T_c = 1.5$ K, the RIXS data point to the critical role of dynamical vertex corrections also for the microscopic description of the superconducting order parameter.

In conclusion, we have presented Ru $L_3$ RIXS measurements of the dynamical response functions in the unconventional superconductor $Sr_2RuO_4$ over a broad range of energy and momentum. We have identified several branches of spin and orbital excitations and revealed the separation of energy scales associated with these two sets of

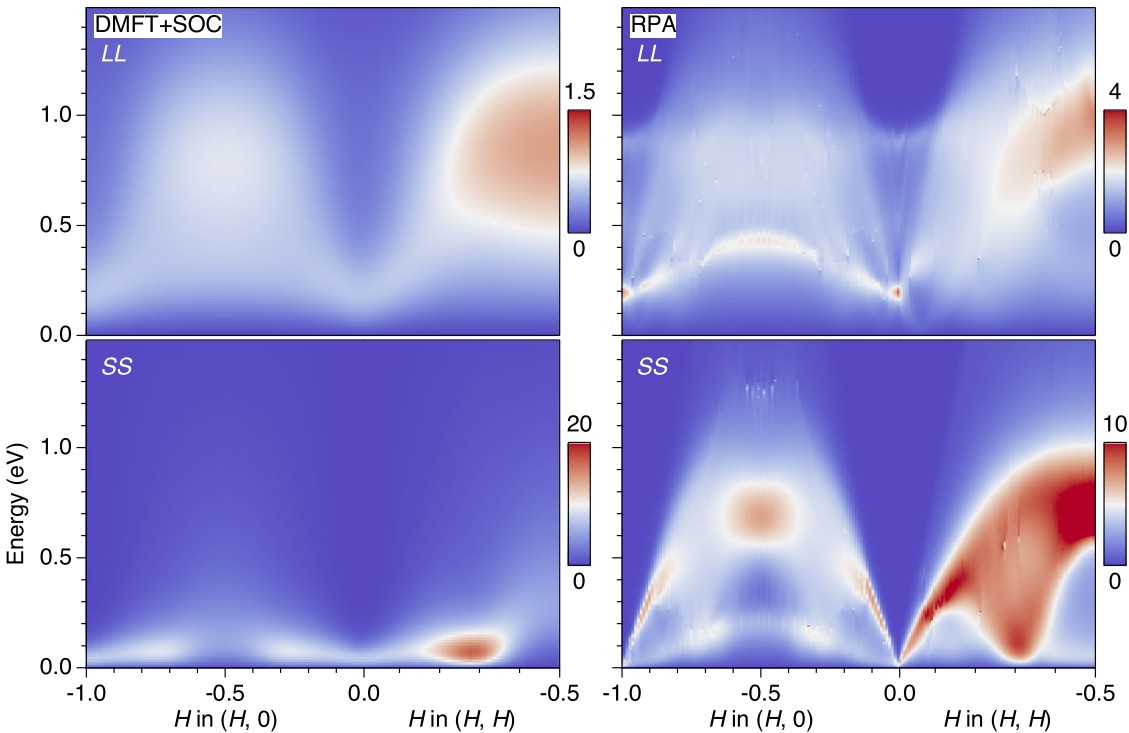

**Fig. 4 | Spin-orbital energy scale separation in DMFT+SOC.** Theoretical orbital ($LL$) and spin ($SS$) susceptibilities obtained by averaging all components, $\frac{1}{3}\sum_\mu X_{L_\mu L_\mu}$ and $\frac{1}{3}\sum_\mu X_{S_\mu S_\mu}$, calculated with DMFT+SOC (left) and RPA (right).

degrees of freedom, a predicted hallmark of Hund metals which had not yet received direct experimental confirmation. The measured spectra are in excellent agreement with theoretical calculations based on DMFT including vertex corrections, while significant discrepancies with the perturbative RPA approximation are found. Our results thus epitomize the power of state-of-the-art many-body theories to yield a detailed, quantitative understanding of complex electronic correlation functions in real materials. By establishing the properties of key collective modes, they also provide a solid baseline for the future identification of the nature and symmetry of SC order in this prominent model compound.

## Methods

### Sample growth and characterization

The $Sr_2RuO_4$ single crystals with superconducting $T_c \sim 1.5$ K were grown by the floating-zone method[39] and pre-aligned using an in-house Laue diffractometer. $Sr_2RuO_4$ has the tetragonal space group $I4/mmm$ with the lattice constants of $a = b = 3.903$ and $c = 12.901$ Å. The in-plane momentum transfers are expressed in the reciprocal lattice units (r.l.u.).

### IRIXS spectrometer

The RIXS experiments were performed using the intermediate-energy RIXS (IRIXS) spectrometer at the P01 beamline of PETRA III at DESY[40]. The incident x-ray energy was tuned to the Ru $L_3$ absorption edge (2838 eV) and incoming photons were monochromatized using a high-resolution monochromator composed of four asymmetrically-cut Si(111) crystals. The polarization of the incident x-ray photons was in the horizontal scattering plane ($\pi$ polarization). The polarizations of the scattered photons were not analyzed. The x-rays were focused to a beam spot of $20 \times 160$ μm² (H × V). Scattered photons from the sample were collected at the scattering angle of 90° (horizontal scattering geometry) using a $SiO_2$ ($10\bar{2}$) ($\Delta E = 60$ meV) diced spherical analyzer with a 1 m arm, equipped with a rectangular [100 (H) × 36 (V)] mm² mask and a CCD camera, both placed in the Rowland geometry.

Collected raw CCD images were transformed into RIXS spectra by summing over the vertical axis of the detector and by binning with 12.5 meV steps along the horizontal axis. To account for the x-ray self-absorption effect, the RIXS intensity was normalized to the total fluorescent intensity collected with an energy-resolved photon detector placed at the scattering angle of 110°. The exact position of the zero energy loss line was determined by measuring non-resonant spectra from silver paint deposited next to the sample. The overall energy resolution of the IRIXS spectrometer at the Ru $L_3$-edge, defined as the full width half maximum of the non-resonant spectrum from silver, was ~80 meV. All the measurements were performed at 25 K (normal state), well above the superconducting $T_c$.

## Data availability

The raw RIXS data generated in this study are available at desycloud: https://desycloud.desy.de/index.php/s/LPt7RJTHqGWLNBD.

## Code availability

The numerical codes used to generate the results in this work are available at desycloud: https://desycloud.desy.de/index.php/s/LPt7RJTHqGWLNBD.

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

## Acknowledgements

We thank A. Damascelli, G. Khaliullin, A. Yaresko, and D. Kukusta for enlightening discussions. We acknowledge DESY (Hamburg, Germany), a member of the Helmholtz Association HGF, for the provision of experimental facilities. The RIXS experiments were carried out at the beamline P01 of PETRA III at DESY. The project was supported by the European Research Council under Advanced Grant No. 669550 (Com4Com) awarded to B.K.. H.S. acknowledges financial support from the JSPS Research Fellowship for Research Abroad and Grants-in-Aid for Scientific Research from JSPS (KAKENHI) (number 22K13994). H.S. and L.W. acknowledge financial support from the Alexander von Humboldt Foundation. S.K. acknowledges financial support by the DFG project HA7277/3-1. H.U.R.S. acknowledges financial support from the ERC synergy grant (854843-FASTCORR). N.K. is supported by KAKENHI (Grant Nos. 18K04715, 21H01033, and 22K19093), Core-to-Core Program (No. JPJSCCA20170002) from JSPS, and a JST-Mirai Program (Grant No. JPMJMI18A3). Research in Dresden benefits from the environment provided by the DFG Cluster of Excellence ct.qmat (EXC 2147, project ID 390858940) awarded to A.P.M.. The Flatiron Institute is a division of the Simons Foundation.

## Author contributions

H.S., L.W., J.B., Z.Y., and H.G. performed the RIXS experiments. M.K., F.J., N.K., and A.P.M. grew the $Sr_2RuO_4$ single crystals. H.S., L.W., and M.K. performed the sample characterizations. H.G. designed the beamline and IRIXS spectrometer. H.S. analyzed the experimental data. H.U.R.S., S.K., N.W., and O.P. developed the computational framework used in the theoretical calculations. S.K., H.U.R.S., A.G., and P.H. carried out the theoretical calculations of the dynamical response functions. H.S. and P.H. constructed the theoretical RIXS intensity from the response functions. H.S., S.K., H.U.R.S., A.G., P.H., and B.K wrote the manuscript with input from all the co-authors. B.K. initiated and supervised the project.

## Funding

## Competing interests

The authors declare no competing interests.
