## [Peer Review File · Nature Communications]

Distinct spin and orbital dynamics in Sr_2RuO_4REVIEWER COMMENTS

Reviewer #1 (Remarks to the Author):

The manuscript by Suzuki et al. presents a comprehensive study of the collective excitations in the normal state of unconventional superconductor Sr₂RuO₄ using Ru L edge resonant inelastic X-ray scattering (RIXS) and dynamical mean-field theory (DMFT) with spin-orbit coupling (SOC) calculations. The authors report distinct spin and orbital excitations that are not observed in inelastic neutron studies. The RIXS data agree well with DMFT+SOC calculations, providing valuable insights into understanding the material's normal-state properties. The experimental data are interesting and up-to-date. However, there are some issues that need to be addressed before I can recommend its publication in Nature Communications.

Firstly, the manuscript focuses on the normal state of Sr₂RuO₄, and it is unclear how these high-energy excitations are relevant to the unconventional superconductivity in Sr₂RuO₄. The authors should provide a clearer discussion of the impact of their results beyond the agreement between RIXS experiment and DMFT calculations. Reference 26 has established that dynamical mean-field theory, incorporating dynamical vertex corrections, outperforms RPA in capturing the spin fluctuations in Sr₂RuO₄. In light of this, it would be beneficial to gain a clearer understanding of how the RIXS results can contribute to the identification of pairing bosons in unconventional superconductivity of Sr₂RuO₄, beyond the acknowledged significance of dynamical vertex correction in DMFT calculations.

Secondly, the authors should discuss the previous O K-edge RIXS study on Sr₂RuO₄ that revealed a nondispersive excitation at ~350 meV, interpreted as holes moving across spin-orbit split t_{2g} states [Physical Review B 91, 155104 (2015)]. This feature should be compared and contrasted with the orbital excitation in the present work, and the reasons for its nondispersive nature at O K-edge and dispersive nature at Ru L-edge should be explained.

Thirdly, the manuscript should provide a discussion on the origin of dispersive orbital excitations in Sr₂RuO₄ and whether this is a fingerprint of Hund metals. The authors should compare and contrast the dispersive orbital excitations in Sr₂RuO₄ with those in Ca₂RuO₄ and Ca₃Ru₂O₇ [Refs. 28, 34], which are non-dispersive and indicate a local nature. The absence of such dispersive excitations in iron-based superconductors, despite being considered Hund's correlated metals, should also be addressed.

Fourthly, the manuscript should explain why the spin excitation is weaker along the [H,0] direction than the [H, H] direction and whether the high-energy spin and orbital excitations are affected by the weakening of low-energy incommensurate spin fluctuations with increasing temperature. The manuscript should also provide a high-temperature measurement, such as at room temperature, for comparison, and it would be more convincing if the softening of spin excitation at q_{ISF} became less evident at room temperature.

Finally, the manuscript should perform fitting procedures to extract peak positions in Figure 2, rather than using global peak maxima, which is a rough method. The authors can follow the procedure employed in Ca₃Ru₂O₇ [Ref. 34] and extract the peak positions using fitting procedures.

Reviewer #2 (Remarks to the Author):

Suzuki et al. reported a Ru L-edge RIXS measurement on Sr₂RuO₄, revealing multiple branches of excitations in energy-momentum space. The manuscript primarily focused on two dispersive branches at low energies, attributed to spin and orbital susceptibility, supported by DMFT+SOC calculations. This agreement provided evidence for spin-orbital separation, a key signature of a Hund's metal.

Sr₂RuO₂ is a classical material for studying unconventional superconductors, exhibiting Fermi-liquid behavior above its superconducting transition temperature. Some theories suggest the system is a Hund metal, which may provide new insights into its electronic properties. The manuscript presents the first Ru L-edge RIXS data of Sr₂RuO₄, which could lend a strong support for this scenario. I found that the data are of high-quality. In addition, the good agreement between data and calculations also supports to the conclusion. However, before I can recommend the publication in Nature Communications, the authors need to clarify the following minor issues:

(1) In Fig. 2a, while the identification of spin excitations is clear for the features near (-0.3, -0.3), the assignment of spin excitations at other momentum positions along the HH-direction from $q = (0, 0)$ to (-0.3, -0.3) is questionable. The spectra lack any clear features and it is not explained how the spin excitations energy is determined. Therefore, the authors need to explain how they determined the positions of spin excitations in the spectra.

(2) The method used to determine the dispersions shown in Fig. 2b is not clear, and the authors need to describe the approach used for these data points, along with the associated error bars.

(3) As Ru is a 4d element, the spin-orbit interaction is expected to be notably stronger than in 3d transition metal oxides. However, the results suggest that spin and orbital should still be treated as separate entities, instead of using the total angular momentum $J = L+S$. Can the authors provide a brief explanation of why J is not a good quantum number in Sr₂RuO₄?

(4) According to theory (e.g. Kugler et al, Ref.19), a key signature of a Hund metal is a significant difference in energy scales between the peaks of spin and orbital susceptibility, differing by at least an order of magnitude. However, the data (e.g. Fig. 2b) suggests that the difference is likely only around three times or smaller, as the spin and orbital components are broad and overlapping in the spectrum (see Fig. 2b). Can the authors briefly comment on the discrepancy between the prediction and the data? Does the smaller energy difference between the putative spin and orbital excitations still qualify Sr₂RuO₄ as a Hund metal?

Reviewer #3 (Remarks to the Author):

The authors reported high-resolution Ru L-edge resonant inelastic x-ray scattering experiment for Sr₂RuO₄. The comprehensive map of the low-energy spin and magnetic excitations is revealed for the first time and is analyzed based on the LDA+DMFT method including vertex corrections. The spin and orbital separation behavior is clear in the RIXS data and is supported by their analysis, together providing a direct evidence of the Hund metal physics in this material. This serves as a crucial important input for the recent hot debate on the superconductivity of this compound. Besides, this work will guide a RIXS route to uncover the Hund metal physics in transition compounds. Thus, I would recommend publication in Nature Communications after the following points are appropriately addressed.

An important observation (with a RIXS experimental support) is that the SOC on Ru 4d does not affect the profile of the dynamical susceptibilities, that is in stark contrast with typical Ru systems

as pointed out by the authors. Could the author reason this difference? Due to local multiplet together with the crystal field or metallic band effect? Explanations or discussions linking the observations with the general Hund physics will be appreciated.

This is probably out of the scope of this work (or model hamiltonian), but can the author comment on the very strong dispersion of the feature C in the RIXS map of Fig1c? Why its maximum is at the (0,0) point and and it gets merged into the crystal field excitation ($t_{2g}^3 e_g^1$) there?

What is the assumption behind the approximation for the RIXS magnetic contribution as $N = S + \alpha * L$? I am a bit surprised that this simple scaling of L allows to fit the dynamical susceptibility with the L-edge RIXS data well. The author mentioned that the analytic form (within fast collision approximation) for t_{2g}^4 system is given in Ref 58. From this perspective, the success of the author's procedure can be reasoned?

The authors excused that the quadrupolar contribution neglected in the present analysis may be an obstacle leading the mismatch between the experiment and theory, especially for the 0.5eV peak around $q=(0,0)$. Since the core-level x-ray excitations are essentially local, a simple atomic-model estimate, for example, for the total RIXS transition operator R gives estimate of the amplitude of the quadrupolar channel with respect to the magnetic one? I am wondering wether the quadrupolar contribution can really be a potential source of this discrepancy between the theory and experiment.

Reply to Reviewer #1

Comment:

The manuscript by Suzuki et al. presents a comprehensive study of the collective excitations in the normal state of unconventional superconductor Sr₂RuO₄ using Ru L edge resonant inelastic X-ray scattering (RIXS) and dynamical mean-field theory (DMFT) with spin-orbit coupling (SOC) calculations. The authors report distinct spin and orbital excitations that are not observed in inelastic neutron studies. The RIXS data agree well with DMFT+SOC calculations, providing valuable insights into understanding the material's normal-state properties. The experimental data are interesting and up-to-date. However, there are some issues that need to be addressed before I can recommend its publication in Nature Communications.

Our Reply:

We appreciate the referee's recognition of the significance of our study in elucidating the spin-orbital separation in the normal-state of Sr₂RuO₄ and in demonstrating the capability of state-of-the-art DMFT+SOC calculations in the precise prediction of the dynamical properties. We have addressed each of the issues raised below.

Comment:

Firstly, the manuscript focuses on the normal state of Sr₂RuO₄, and it is unclear how these high-energy excitations are relevant to the unconventional superconductivity in Sr₂RuO₄. The authors should provide a clearer discussion of the impact of their results beyond the agreement between RIXS experiment and DMFT calculations. Reference 26 has established that dynamical mean-field theory, incorporating dynamical vertex corrections, outperforms RPA in capturing the spin fluctuations in Sr₂RuO₄. In light of this, it would be beneficial to gain a clearer understanding of how the RIXS results can contribute to the identification of pairing bosons in unconventional superconductivity of Sr₂RuO₄, beyond the acknowledged significance of dynamical vertex correction in DMFT calculations.

Our Reply:

We thank Reviewer #1 for prompting us to clarify the potential impact of our study on the understanding of the pairing bosons for unconventional superconductivity in Sr₂RuO₄. The RIXS data and the corresponding DMFT+SOC results provide a comprehensive description of the momentum distribution of the low-energy magnetic excitations that are being intensely discussed as possible pairing bosons for unconventional superconductivity (Fig. 3). In particular, the intensity balance between incommensurate excitations and excitations around $\mathbf{q} = 0$ is important for the relative stability of different pairing symmetries [see e.g. Rømer et al., Phys. Rev. Research 4, 033011 (2022) and references therein]. In response to the Reviewer's query (as well as a related point raised by Reviewer 2, see below), we have also added a new analysis of the spin-orbit composition of the low-energy magnetic excitations (Extended Data Fig. 8), which demonstrates that the spin-orbit interaction does have a non-negligible influence, although these excitations are mostly of spin character. This information is generally relevant for quantitative models of the magnetically mediated pairing mechanisms, and for relative stability of different, nearly degenerate pairing wavefunctions in the framework of such models.

We further show that many of the salient features of these excitations are not captured correctly by RPA calculations; the theoretical results of Ref. [31] are a case in point.

To further clarify these implications, we have revised the corresponding paragraph on page 10 as follows:

Our combined RIXS and DMFT+SOC results provide a comprehensive description of the momentum distribution, dispersion relation, and spin-orbit composition of low-energy magnetic excitations, which can serve as crucial input for approximate solutions of the Eliashberg equations. Recent theoretical studies suggest that static (RPA) and dynamic (DMFT) vertex approximations lead to qualitatively different SC ground states [31, 34-36]. Although computational challenges prohibit rigorous extrapolation of our theoretical results to low temperatures near $T_c = 1.5$ K, the RIXS data point to the critical role of dynamical vertex corrections also for the microscopic description of the superconducting order parameter.

Comment:

Secondly, the authors should discuss the previous O K-edge RIXS study on Sr₂RuO₄ that revealed a nondispersive excitation at ~350 meV, interpreted as holes moving across spin-orbit split t_{2g} states [Physical Review B 91, 155104 (2015)]. This feature should be compared and contrasted with the orbital excitation in the present work, and the reasons for its nondispersive nature at O K-edge and dispersive nature at Ru L-edge should be explained.

Our Reply:

The O K-edge study mentioned by the referee is relevant because it reports orbital excitations in an energy range that is compatible with the one covered by our Ru L-edge experiments. We have therefore added a reference to this work to the revised manuscript (Ref. [28]), with the sentence “The energy scale of orbital fluctuations near the zone center agrees with that of the O K-edge RIXS data [28].”. However, due to intrinsic limitations of O K-edge RIXS, this study only covers a very limited range of momenta along a single reciprocal-space axis [(H, 0) with $H \leq 0.21$]. The absence of a clearly discernible dispersion in this energy range is not inconsistent with our much more extensive data, and does not warrant a detailed discussion in our view.

More generally, we note that the excellent agreement between the Ru L-edge RIXS data and the DMFT+SOC calculations over a wide range of momentum space, supplemented by the group-theoretical analysis presented in *Methods*, provides conclusive evidence that Ru L-edge RIXS cross section directly reflects the magnetic correlation functions of Sr₂RuO₄, which encompass dispersive branches in the (\mathbf{q} , ω) space. In contrast, O K-edge RIXS probes the properties of the Ru 4d electrons only via their hybridization with the O 2p orbitals. Consequently, the O K-edge RIXS cross section shows a large deviation from the magnetic dynamical correlation functions of the system. This is one of our motivations to develop the IRIXS spectrometer dedicated to the tender x-ray region comprising the Ru L-edges.

Comment:

Thirdly, the manuscript should provide a discussion on the origin of dispersive orbital excitations in Sr₂RuO₄ and whether this is a fingerprint of Hund metals. The authors should compare and contrast the dispersive orbital excitations in Sr₂RuO₄ with those in Ca₂RuO₄ and Ca₃Ru₂O₇ [Refs. 28, 34], which are non-dispersive and indicate a local nature. The absence of such dispersive excitations in iron-based superconductors, despite being considered Hund's correlated metals, should also be addressed.

Our Reply:

As we have emphasized in the title and manuscript, the most direct fingerprint of the Hund metal Fermi-liquid state in Sr₂RuO₄ is the energy separation of spin and orbital fluctuations, as clearly revealed by our RIXS data and DMFT+SOC calculations (Fig. 2a, b and Fig. 4). The dispersion of orbital excitations is a more general phenomenon not directly related to the Hund-metal concept. It originates from the mobility of orbital excitations due to the large bandwidth of Sr₂RuO₄ with a tetragonal crystal structure with a Ru-O-Ru bond angle of 180°. In contrast, the nondispersive orbital excitations revealed in our previous work on Ca₂RuO₄ and Ca₃Ru₂O₇ indicate that the orbital excitations cannot freely move to the neighboring sites, as the rotation of RuO₆ octahedra reduces the bond angle and hopping integrals.

To convey these considerations, we have added the following sentences in the paragraph explaining the high-energy orbital excitations:

*“The dispersion of orbital fluctuations originates from the large bandwidth of Sr₂RuO₄ with a tetragonal crystal structure. In contrast, the nondispersive orbital excitations observed in orthorhombic Ca₂RuO₄ [26] and Ca₃Ru₂O₇ [27] indicate that the local *dd* excitations cannot freely move to the neighboring sites, as the rotation of the RuO₆ octahedra significantly reduces the hopping integrals.”*

We do not regard the Reviewer's statement regarding “the absence of such dispersive excitations in iron-based superconductors” as an established experimental fact. High-energy RIXS spectra of iron-based superconductors indeed exhibit a large continuum without notable dispersion [see, e.g., K. Zhou et al., Nat. Commun. **4**, 1470 (2013)], but as the fermiology of the iron-based superconductors is more complex than that of Sr₂RuO₄, it is quite possible that the continuum is composed of several branches of dispersive modes. To the best of our knowledge, a comprehensive investigation of these excitations in momentum space has not been reported in the literature. In addition, we emphasize again that the Hund metal nature and the dispersion of orbital excitations are distinct – and not causally related – findings of our study.

Comment:

Fourthly, the manuscript should explain why the spin excitation is weaker along the [H,0] direction than the [H, H] direction and whether the high-energy spin and orbital excitations are affected by the weakening of low-energy incommensurate spin fluctuations with increasing temperature. The manuscript should also provide a high-temperature measurement, such as at

room temperature, for comparison, and it would be more convincing if the softening of spin excitation at q_{ISF} became less evident at room temperature.

Our Reply:

The relative intensity of spin excitations and the location of the incommensurate spin fluctuation are determined by the nesting conditions between the multiple Fermi surface sheets of Sr_2RuO_4 . It is well established that the nesting between the α and β sheets with $q_{\text{ISF}} = (0.3, 0.3)$, as demonstrated by ARPES Fermi surface measurements (Tamai et al., Ref. 25), drives the incommensurate spin fluctuations. In contrast, the nesting is only partial along the $(H, 0)$ direction, resulting in reduced intensity of the spin fluctuations in that direction. **These considerations are now included in the revised manuscript (see the paragraph discussing Fig. 1c on page 5).**

Our RIXS study has yielded a comprehensive description of the spin and orbital correlation functions of Sr_2RuO_4 at low temperature, and a detailed comparison to state-of-the-art many-body theory. At the temperature chosen for the experiments (25 K), the key result of our study (namely the spin-orbital separation) is clearly apparent. The question raised by the Reviewer about the robustness of the spin-orbital separation with increasing temperature is interesting, but will require many more synchrotron measurements and DMFT+SOC calculations to establish the phenomenology, the underlying mechanisms, and the relationship to transport and photoemission data. This subject will therefore require dedicated study and cannot be covered by the current publication.

Comment:

Finally, the manuscript should perform fitting procedures to extract peak positions in Figure 2, rather than using global peak maxima, which is a rough method. The authors can follow the procedure employed in $\text{Ca}_3\text{Ru}_2\text{O}_7$ [Ref. 34] and extract the peak positions using fitting procedures.

Our Reply:

Fitting to model functions such as damped-harmonic-oscillator or Voigt profiles is sensible and commonly practiced for spectral features arising from transitions between sharply defined intra-atomic multiplets or low-energy collective modes such as magnons or phonons. Following standard practice, we have performed such fitting routines in our previous Ru L -edge RIXS publications on Mott insulating Ru compounds whose excitation spectra can be neatly decomposed into superpositions of single-peak features [Suzuki et al. Nat. Mater. **18**, 563 (2019), Gretarsson et al. Phys. Rev. B **100**, 045123 (2019), Suzuki et al. Nat. Commun. **12** 4512 (2021), Takahashi et al., Phys. Rev. Lett. **127**, 227201 (2021)]. In our publication on metallic $\text{Ca}_3\text{Ru}_2\text{O}_7$ mentioned by the Referee [Bertinshaw et al., Phys. Rev. B **103**, 085108 (2021)], we have only employed this procedure to extract the energies of low-energy magnon features in the antiferromagnetically ordered state, which are again amenable to a description in terms of single-peak profiles.

The excitation continua in metallic systems without magnetic order such as Sr_2RuO_4 (and $\text{Ca}_3\text{Ru}_2\text{O}_7$ at energies above the magnon band) arise from a complex convolution of two-particle excitations at different momenta, and there is no theoretical basis for a decomposition into single-peak profiles. This has been one of the primary motivations for comparing our experimental results to the outcome of numerical DMFT+SOC calculations, which is also not amenable to a simple decomposition into DHO or Voigt profiles, as shown in Extended Data Fig. S3. As we have demonstrated, RIXS and DMFT+SOC spectra are in excellent agreement, after adjusting the latter results with a single parameter, α , that takes into account the complexities of the RIXS cross section.

To facilitate comparison to phenomenological calculations and data from other spectroscopic probes [see for instance calculations performed in the cuprate superconductors, Le Tacon et al., Nature Physics 7, 725 (2011)], we have additionally plotted the intensity maxima of the spin and orbital correlations in Fig. 2. Rather than fitting to model profiles without theoretical justification, we determined the maxima via a second-derivative analysis, which captures the local curvature of spectral features. This analysis scheme does not require any assumptions is widely accepted in the analysis of angle-resolved photoemission data, where multiple band dispersions are close in energy. The dispersion plot of spin fluctuations in Fig. 2 thus relies on a straightforward and transparent method and is useful in our view, but we are prepared to abandon it for the sake of publication, if referees and editors insist.

To convey the above considerations, we have included a new section “Second derivative of RIXS intensity” in Methods, together with a new Extended Data Fig. 2. Furthermore, we have changed the title of a section in Methods, from “Simulation of RIXS intensity” to “Fitting of RIXS intensity based on theoretical susceptibilities”.

Reply to Reviewer #2

Comment:

Suzuki et al. reported a Ru L-edge RIXS measurement on Sr_2RuO_4 , revealing multiple branches of excitations in energy-momentum space. The manuscript primarily focused on two dispersive branches at low energies, attributed to spin and orbital susceptibility, supported by DMFT+SOC calculations. This agreement provided evidence for spin-orbital separation, a key signature of a Hund's metal.

Sr_2RuO_2 is a classical material for studying unconventional superconductors, exhibiting Fermi-liquid behavior above its superconducting transition temperature. Some theories suggest the system is a Hund metal, which may provide new insights into its electronic properties. The manuscript presents the first Ru L-edge RIXS data of Sr_2RuO_4 , which could lend a strong support for this scenario. I found that the data are of high-quality. In addition, the good agreement between data and calculations also supports to the conclusion. However, before I can recommend the publication in Nature Communications, the authors need to clarify the following minor issues:

Our Reply:

We thank Reviewer #2 for supporting the acceptance of our manuscript and its significance as a model case to approach the modern questions of magnetism using RIXS. In accordance with the Referee's comments, we have revised our manuscript as detailed below.

Comment:

(1) In Fig. 2a, while the identification of spin excitations is clear for the features near (-0.3, -0.3), the assignment of spin excitations at other momentum positions along the HH-direction from $q = (0, 0)$ to (-0.3, -0.3) is questionable. The spectra lack any clear features and it is not explained how the spin excitations energy is determined. Therefore, the authors need to explain how they determined the positions of spin excitations in the spectra.

Our Reply:

We thank Reviewer #2 for raising this issue. We have used the maximum curvature from second derivative plot to determine the dispersion of the spin excitations. We have included a new section "Second derivative of RIXS intensity" in Methods, together with a new Extended Data Fig. 2. Furthermore, we have changed the title of a section in Methods, from "Simulation of RIXS intensity" to "Fitting of RIXS intensity based on theoretical susceptibilities". See also our reply to the last question from Reviewer #1 for more details.

Comment:

(2) The method used to determine the dispersions shown in Fig. 2b is not clear, and the authors need to describe the approach used for these data points, along with the associated error bars.

Our Reply:

The dispersion of orbital fluctuations is determined based on the global peak maxima of the RIXS spectra, while the that of spin fluctuations from the maximum curvature deduced from the second-derivative analysis. The error bars associated with the maximum search algorithm mostly concern the number of smoothing of the data, but they are typically much smaller than the marker size. The definition of the dispersions is included in the main text:

Here, the dispersion of the orbital fluctuations is defined as the global peak maxima of the RIXS spectra, and that of the spin fluctuations as the local maxima of spectral curvature deduced from the second derivative analysis (Expanded Data Fig. 2).

The estimation of the error bars is also added in the caption of Fig. 2:

The dispersion of the spin fluctuations is defined as the maxima of local curvature deduced from the second derivative analysis, and that of orbital fluctuations as the global maxima of the original RIXS spectra. The typical error bars associated with the numerical maximum search are smaller than the marker size.

Comment:

(3) As Ru is a 4d element, the spin-orbit interaction is expected to be notably stronger than in 3d transition metal oxides. However, the results suggest that spin and orbital should still be

treated as separate entities, instead of using the total angular momentum $J = L+S$. Can the authors provide a brief explanation of why J is not a good quantum number in SrRuO₄?

Our Reply:

While the t_{2g} electrons carry orbital angular momentum, the finite bandwidth of the itinerant electrons and the tetragonal distortion of the local crystal field partially quench the orbital momentum. Nonetheless, the SOC brings about significant modification of the single-particle band structure at certain high-symmetry \mathbf{k} points in the Brillouin zone, when multiple bands are degenerate in energy. It is well known that in Sr₂RuO₄ this degeneracy occurs in the diagonal direction, which leads to the separation of the Fermi-surface sheets, as initially pointed out by Ref. 33 in the revised draft. However, the impact on the local (i.e. \mathbf{k} -integrated) moment is negligibly small.

Yet, we appreciate the Reviewer #2's viewpoint that our work provides a model example where the itinerancy of the t_{2g} electrons reconstructs the local spin-orbital J physics found in Ru-based Mott insulators. To further clarify the effect of SOC in the dynamical properties of Sr₂RuO₄, we added a difference plot (DMFT+SOC - DMFT) as Extended Data Fig. 8, which reveals a significant modification of the spin susceptibilities at $\mathbf{q} = (0, 0)$ and at \mathbf{q}_{ISF} , as well as a minor change in the orbital susceptibilities. Additionally, we have included the above considerations in the corresponding paragraph in the main text:

This situation is contrasted to the spin-orbital J physics in the Mott insulating counterpart Ca₂RuO₄ [26] and the cubic K₂RuCl₆ [32], whose magnetic ground states are determined by the interplay between the ionic J multiplets and the strength of intersite exchange interactions. While the t_{2g} electrons of Sr₂RuO₄ carry orbital angular momentum, the finite bandwidth of the itinerant electrons partially quenches the orbital momentum. Nonetheless, the SOC brings about significant modification of the single-particle band structure at certain high-symmetry momenta in the Brillouin zone, when multiple bands are degenerate in energy. It is well known that in Sr₂RuO₄ this degeneracy occurs in the diagonal direction, which leads to the separation of the Fermi-surface sheets [25,33]. Correspondingly, the effect of SOC on the dynamical susceptibilities is most pronounced in the spin fluctuations at $\mathbf{q} = (0, 0)$ and at \mathbf{q}_{ISF} , while the effect on the orbital fluctuations remains minor (Extended Data Fig. 8).

Comment:

(4) According to theory (e.g. Kugler et al, Ref.19), a key signature of a Hund metal is a significant difference in energy scales between the peaks of spin and orbital susceptibility, differing by at least an order of magnitude. However, the data (e.g. Fig. 2b) suggests that the difference is likely only around three times or smaller, as the spin and orbital components are broad and overlapping in the spectrum (see Fig. 2b). Can the authors briefly comment on the discrepancy between the prediction and the data? Does the smaller energy difference between the putative spin and orbital excitations still qualify Sr₂RuO₄ as a Hund metal?

Our Reply:

We first stress that our DMFT calculations were performed for the same model as that in Kugler et al. and indeed show a clear energy scale separation: the spin fluctuation is peaked at ~ 0.06 eV and the orbital fluctuations are broadly peaked around 0.8 - 1 eV (See Extended Data Fig.3). Note also that the dispersion plotted in Fig. 2b does not consider the spectral weight. The RIXS spectral weight of the orbital fluctuations is maximal at $\mathbf{q} = (-0.5, -0.5)$ around 0.55 eV, which is indeed one order of magnitude larger than the spin fluctuation peak energy of 0.06 eV.

As the Reviewer #2 correctly pointed out, the spin fluctuations overlap with the low-energy tail of the orbital fluctuations that have dominant spectral weight. This apparent discrepancy from the “spin-orbital separation” concept is primarily attributed to the difference between the RIXS cross section and the dynamical susceptibilities. Note that in constructing the theoretical RIXS intensity from the dynamical susceptibilities, we have assumed the relation $\mathbf{N} = \mathbf{S} + \alpha\mathbf{L}$ for the magnetic transition operator [Eq. (5) of *Methods*]. For the DMFT calculations, the fitting parameter is α set to 3.6, which enhances the contribution from the orbital correlation functions by a factor of 13. This is due to the reduced overall intensity scale of the orbital susceptibilities compared to the spin susceptibilities (see the color bars in Extended Data Fig. 3). Therefore, the actual spectral weight overlap is quite small, which validates the separation of spin and orbital excitations.

To convey this consideration to the reader, we have added the following sentence in the corresponding chapter in Methods:

Note that $\alpha = 3.6$ for DMFT+SOC and DMFT enhances the contributions from the orbital susceptibilities by a factor of ~ 13 . This is due to the reduced intensity scale of the orbital susceptibilities compared to the spin susceptibilities (see the color bars in Extended Data Fig. 3).

Reply to Reviewer #3

Comment:

The authors reported high-resolution Ru L-edge resonant inelastic x-ray scattering experiment for Sr₂RuO₄. The comprehensive map of the low-energy spin and magnetic excitations is revealed for the first time and is analyzed based on the LDA+DMFT method including vertex corrections. The spin and orbital separation behavior is clear in the RIXS data and is supported by their analysis, together providing a direct evidence of the Hund metal physics in this material. This serves as a crucial important input for the recent hot debate on the superconductivity of this compound. Besides, this work will guide a RIXS route to uncover the Hund metal physics in transition compounds. Thus, I would recommend publication in Nature Communications after the following points are appropriately addressed.

Our Reply:

We are grateful to Reviewer #3 for recognizing the importance of our work and his/her recommendation of publication.

Comment:

An important observation (with a RIXS experimental support) is that the SOC on Ru 4d does not affect the profile of the dynamical susceptibilities, that is in stark contrast with typical Ru systems as pointed out by the authors. Could the author reason this difference? Due to local multiplet together with the crystal field or metallic band effect? Explanations or discussions linking the observations with the general Hund physics will be appreciated.

Our Reply:

We appreciate Reviewer #3 for raising the issue. As we have explained above in the reply to Reviewer #2, we primarily ascribe the different effect of SOC to the itinerancy of the t_{2g} electrons. The revised manuscript in page 10 provides the answer to this comment.

Comment:

This is probably out of the scope of this work (or model hamiltonian), but can the author comment on the very strong dispersion of the feature C in the RIXS map of Fig1c? Why its maximum is at the (0,0) point and and it gets merged into the crystal field excitation ($t_{2g}^3 e_g^1$) there?

Our Reply:

As the Reviewer #3 correctly understands, the detailed theoretical description of the features B and C involving the e_g orbitals is beyond the scope of this work. This is because one of the key findings of this work is that the established t_{2g} model of Sr_2RuO_4 provides excellent description of the dynamical properties up to ~ 1.5 eV, without the need of fine-tuning of the parameters. Inclusion of e_g orbitals to our model Hamiltonian would require comprehensive consistency checks with existing experimental results including ARPES and INS, which deviates from the main line of argument of this manuscript.

Nevertheless, we share the Reviewer #3's opinion that a discussion concerning the dispersive features B and C holds significance and some statements should be included in our manuscript. At the high-symmetry $\mathbf{q} = (0, 0)$ point, the dd excitations to the $d_{x^2-y^2}$ and $d_{3z^2-r^2}$ orbitals remain localized and are almost degenerate in energy under the small tetragonal distortion of RuO_6 octahedra. At finite in-plane \mathbf{q} 's, the excitations to the planar $d_{x^2-y^2}$ orbitals show energy dispersion due to large overlap integrals with the planar O $2p$ orbitals, while those to the out-of-plane $d_{3z^2-r^2}$ orbitals have little in-plane dispersion due to the small overlap integrals. The nondispersive feature B and dispersive feature C are thus primarily ascribed to the transitions to the $d_{3z^2-r^2}$ and $d_{x^2-y^2}$ orbitals, respectively. However, this assignment remains only qualitative as three t_{2g} bands form the metallic ground state, and the features B and C stand out only weakly as part of a Slater continuum above ~ 1 eV. This situation is contrasted to the orbital excitations in quasi-one-dimensional Mott insulator Sr_2CuO_3 [J. Schlappa et al. Nature **485** 82 (2012)], where the orbital excitations from the single $d_{x^2-y^2}$ orbital form dispersive sharp peaks in the RIXS spectra. The full description of dispersion relation of the feature C requires a

dedicated theoretical modeling and computation, and we refrain from a definitive statement on why it is maximal at the (0, 0) point.

To convey this qualitative consideration, we have deleted the sentence “Considering...” in the previous version and added following sentences in the corresponding paragraph in page 6:

At the high-symmetry (0, 0) point, the dd excitations to the $d_{x^2-y^2}$ and $d_{3z^2-r^2}$ orbitals remain localized and are almost degenerate in energy under the small tetragonal distortion of the RuO_6 octahedra. At finite in-plane q 's, the excitations to the planar $d_{x^2-y^2}$ orbitals show energy dispersion due to large overlap integrals with the planar O $2p$ orbitals, while those to the out-of-plane $d_{3z^2-r^2}$ orbitals have little in-plane dispersion due to the small overlap integrals. The nondispersive feature B and dispersive feature C are thus primarily ascribed to the transitions to the $d_{3z^2-r^2}$ and $d_{x^2-y^2}$ orbitals, respectively.

Comment:

What is the assumption behind the approximation for the RIXS magnetic contribution as $N = S + \alpha L$? I am a bit surprised that this simple scaling of L allows to fit the dynamical susceptibility with the L-edge RIXS data well. The author mentioned that the analytic form (within fast collision approximation) for t_{2g}^4 system is given in Ref 58. From this perspective, the success of the author's procedure can be reasoned?

Our Reply:

Our approach is to construct the effective RIXS operator that is valid within the spin-orbital manifold of $S = L = 1$. Our approximation $N = S + \alpha L$ is the simplest linear combination of L and S that transforms as a pseudovector. On the other hand, the analytic expression for the magnetic transition operator $N = (N_x, N_y, N_z)$ for an t_{2g}^4 ion derived in Kim et al. (Ref. 62 in the revised draft) is given by (apart from a constant prefactor): $N_z = 2L_z - 4L_z^2 S_z + L_z(L \cdot S) + (L \cdot S)L_z$. N_x and N_y follow from symmetry. While the L operator appears as it is (first term), the S operator is coupled with quadratic terms of L operators (last three terms). In Sr_2RuO_4 , the spin and orbital fluctuations are energetically separated (Extended Data Fig. 3). Therefore, spin excitation intensity is suppressed in the RIXS cross section. Correspondingly, we need to enhance the L term in our fit ($\alpha = 3.6$). This reasoning is added in the corresponding section in *Methods* (please see page 19).

Comment:

The authors excused that the quadrupolar contribution neglected in the present analysis may be an obstacle leading the mismatch between the experiment and theory, especially for the 0.5eV peak around $q=(0,0)$. Since the core-level x-ray excitations are essentially local, a simple atomic-model estimate, for example, for the total RIXS transition operator R gives estimate of the amplitude of the quadrupolar channel with respect to the magnetic one? I am wondering whether the quadrupolar contribution can really be a potential source of this discrepancy between the theory and experiment.

Our Reply:

The analytic expressions for quadrupolar transition operators are also listed in Ref. 58. The transition amplitude depends strongly on the orbital wavefunctions of the ground state and excited states, so a general statement about the amplitude is not possible. However, we may provide a rough estimate based on our previous RIXS study of the cubic t_{2g}^4 Ru-based Mott insulator K_2RuCl_6 (Ref. [32]). In this compound, the $J = 0$ nonmagnetic ground state belongs to the A_{1g} representation of the O_h point group. Therefore, the irreducible representations of the transition operator are directly connected to those of the final states. While the main peak is given by the magnetic transitions to the $J = 1$ (T_{1g}) states, the quadrupolar (T_{2g} and E_g) transitions to the $J = 2$ and $(S, L) = (0, 2)$ states are also clearly identified. The intensity of the $J = 2$ transitions is about half of the $J = 1$ transitions. The presence of the quadrupolar transition is thus an established experimental fact and should not be discarded in the interpretation of the experimental data.

Figure 1 Ru L_3 RIXS spectra of K_2RuCl_6 [32].

To convey this consideration, we have added a sentence on the intensity of the quadrupolar transitions in the corresponding section in *Methods*:

The intensity of the $J = 2$ transitions is about half of that of the $J = 1$ transitions.

Additional Code description

To provide a description of the functionality of our data production routines, we have added the following statement in *Methods*:

Collected raw CCD images were transformed into RIXS spectra by summing over the vertical axis of the detector and by binning with 12.5 meV steps along the horizontal axis.

REVIEWER COMMENTS

Reviewer #1 (Remarks to the Author):

Suzuki et al. have addressed most of my previous comments in the response letter and the revised manuscript. I appreciate that the authors have clarified the origin of the dispersive orbital excitations in Sr₂RuO₄ and presented compelling evidence of spin-orbital separation in Hund metals. I recommend its publication in Nature Communications if the authors clarify the following issues:

1. The authors state, "It is well established that the nesting between the α and β sheets with q ISF, as demonstrated by angle-resolved photoemission measurements [25], drives the incommensurate spin fluctuations with q ISF." Can the authors provide more details and relevant references to explain why the nesting between the α and β sheets drives spin fluctuations? Reference [25] reports the impact of spin-orbit coupling in splitting β and γ sheets and does not serve this purpose.
2. The authors provide the second derivative plot of the RIXS intensity (extended data Fig. 2) to determine the dispersions of spin and orbital fluctuations. In the (H, H) direction, the markers follow the maxima of the second derivative. However, in the (H, 0) direction, there appears to be an additional dispersion below the orbital fluctuation with notably higher intensity. It disperses from ~ 0.2 eV at (0,0) to ~ 0.3 eV at (-0.5,0). Can the authors comment on this extra dispersion?

Reviewer #2 (Remarks to the Author):

This is my second review of the manuscript by Suzuki et al. The authors have addressed most of my concerns. Although using the maximum and second derivative in the spectrum to determine the dispersion in orbital and spin excitations is not ideal, it appears to be the most feasible and objective way for the current set of data. The authors have also provided clear arguments for my questions and have revised the manuscript accordingly. I do not have any additional comments and can recommend the manuscript for publication in Nature Communications.

Reviewer #3 (Remarks to the Author):

I have read the authors' responses and revised manuscript. The authors addressed all my concerns sufficiently, and the manuscript has been improved largely. I recommend the publication of this great work in Nature Communications.

Reply to Reviewer #1

Comment:

Suzuki et al. have addressed most of my previous comments in the response letter and the revised manuscript. I appreciate that the authors have clarified the origin of the dispersive orbital excitations in Sr₂RuO₄ and presented compelling evidence of spin-orbital separation in Hund metals. I recommend its publication in Nature Communications if the authors clarify the following issues:

Our Reply:

We thank Reviewer #1 for recognizing the thoroughness of our revisions. Below, we provide detailed responses to his/her additional requests.

Comment:

The authors state, "It is well established that the nesting between the α and β sheets with q_{SF} , as demonstrated by angle-resolved photoemission measurements [25], drives the incommensurate spin fluctuations with q_{SF} ." Can the authors provide more details and relevant references to explain why the nesting between the α and β sheets drives spin fluctuations? Reference [25] reports the impact of spin-orbit coupling in splitting β and γ sheets and does not serve this purpose.

Our Reply:

We thank Reviewer #1 for pointing out that citing former Ref.25 is irrelevant here as the nesting between the two Fermi surface sheets has been discussed in earlier stages of Sr₂RuO₄ research. In the revised draft, we cite Damascelli et al., PRL **85** 5195 (2020) and I. I. Mazin et al., PRL **82** 4324 (1999). We do not believe that additional explanation is necessary as the statement here refers to textbook physics of the enhancement of the spin susceptibility by Fermi surface nesting.

Comment:

The authors provide the second derivative plot of the RIXS intensity (extended data Fig. 2) to determine the dispersions of spin and orbital fluctuations. In the (H, H) direction, the markers follow the maxima of the second derivative. However, in the (H, 0) direction, there appears to be an additional dispersion below the orbital fluctuation with notably higher intensity. It disperses from ~ 0.2 eV at (0,0) to ~ 0.3 eV at (-0.5,0). Can the authors comment on this extra dispersion?

Our Reply:

We thank Reviewer #1 for pointing out this important observation. This dispersion inferred from the curvature maximum along the (H, 0) direction matches well with the $\langle L_z L_z \rangle$ orbital susceptibility in our DMFT+SOC calculation. We included the following sentence in Methods:

A weak orbital fluctuation branch which disperses from ~ 0.2 eV at $(0,0)$ to ~ 0.3 eV at $(-0.5,0)$ is also identified, which corresponds to the $\langle L_z L_z \rangle$ component of the orbital dynamical response function (see Extended Data Fig. 3).

REVIEWERS' COMMENTS

Reviewer #1 (Remarks to the Author):

The authors have clarified my remaining concerns. I recommend its publication in Nature Com

Reply to Reviewer #1

Comment:

The authors have clarified my remaining concerns. I recommend its publication in Nature Communications.

Our Reply:

We thank Reviewer #1 for appreciating our revision and his/her recommendation of publication in Nature Communications.